# A Lesion-Based Convolutional Neural Network Improves Endoscopic Detection and Depth Prediction of Early Gastric Cancer

**DOI:** 10.3390/jcm8091310

**Published:** 2019-08-26

**Authors:** Hong Jin Yoon, Seunghyup Kim, Jie-Hyun Kim, Ji-Soo Keum, Sang-Il Oh, Junik Jo, Jaeyoung Chun, Young Hoon Youn, Hyojin Park, In Gyu Kwon, Seung Ho Choi, Sung Hoon Noh

**Affiliations:** 1Department of Internal Medicine, Gangnam Severance Hospital, Yonsei University College of Medicine, Seoul 06273, Korea; 2SELVAS AI Inc., Seoul 08594, Korea; 3Department of Surgery, Gangnam Severance Hospital, Yonsei University College of Medicine, Seoul 06273, Korea

**Keywords:** early gastric cancer, artificial intelligence, convolutional neural networks, endoscopy

## Abstract

In early gastric cancer (EGC), tumor invasion depth is an important factor for determining the treatment method. However, as endoscopic ultrasonography has limitations when measuring the exact depth in a clinical setting as endoscopists often depend on gross findings and personal experience. The present study aimed to develop a model optimized for EGC detection and depth prediction, and we investigated factors affecting artificial intelligence (AI) diagnosis. We employed a visual geometry group(VGG)-16 model for the classification of endoscopic images as EGC (T1a or T1b) or non-EGC. To induce the model to activate EGC regions during training, we proposed a novel loss function that simultaneously measured classification and localization errors. We experimented with 11,539 endoscopic images (896 T1a-EGC, 809 T1b-EGC, and 9834 non-EGC). The areas under the curves of receiver operating characteristic curves for EGC detection and depth prediction were 0.981 and 0.851, respectively. Among the factors affecting AI prediction of tumor depth, only histologic differentiation was significantly associated, where undifferentiated-type histology exhibited a lower AI accuracy. Thus, the lesion-based model is an appropriate training method for AI in EGC. However, further improvements and validation are required, especially for undifferentiated-type histology.

## 1. Introduction

Accurate staging is the basis for determining an appropriate treatment plan for suspected early gastric cancer (EGC) based on endoscopy or biopsy findings. As the indications for endoscopic resection (ER) and minimally invasive surgery are usually decided by the T-stage, tumor invasion depth is crucial for determining the treatment modality [1,2,3].

EGC is categorized as tumor invasion of the mucosa (T1a) or that of the submucosa (T1b). Endoscopic ultrasonography (EUS) is useful for T-staging of gastric cancer because it can delineate each gastric wall layer [4,5]. However, EUS is not superior to conventional endoscopy for T-staging of EGC, having a low accuracy of approximately 70% [6,7]. Therefore, there has been increasing interest in the field of medical imaging regarding modalities for predicting EGC depth.

Recently, deep learning-based artificial intelligence (AI) has shown remarkable progress across multiple medical fields. Diagnostic imaging is currently the highest and most efficient application of AI-based analyses in medical fields [8,9]. AI using endoscopic images has been applied to diagnose neoplasms in the gastrointestinal tract [10,11]. Deep convolutional neural networks (CNNs) are a type of deep learning model that are widely used for image analysis [12]. However, they differ from general image classification as the difference in EGC depth in endoscopic images is subtler and more difficult to discern. Therefore, more sophisticated image classification methods are required.

Although conditions such as easily distinguishable visual features and large-scale datasets play key roles in the performance improvements of natural image classification models, these conditions are difficult to be applied to EGC detection and EGC depth prediction models. Although the definitions of invasion depth in EGC differ, features such as textures, shapes, and colors are visually similar. In addition, each degree of invasion depth may not have sufficient training images to cover all types of visual features, because of the fine-scale granularity. Therefore, models for EGC detection and depth prediction may be used to focus on other visually distinguishable patterns rather than EGC. For example, model weights may initially be tuned to find a tiny particle appearing on most images in a T1b-EGC training set rather than extracting features from homogeneous regions. Therefore, it is critical to guide the model to learn the visual features of EGC regions rather than those of other gastric textures.

The present study aims to develop a model and training method optimized for EGC depth prediction, evaluate its diagnostic performance, and investigate factors affecting AI diagnosis.

## 2. Patients and Methods

### 2.1. Patients

This study included 800 patients (538 men and 262 women; age: 26–92 years; mean age: 62.6 years) with an endoscopic diagnosis of EGC at the Gangnam Severance Hospital, Yonsei University College of Medicine, Seoul, Korea, between January 2012 and March 2018. EGC was suspected based on endoscopy findings and all patients underwent a curative treatment by either operation or ER for gastric cancer. The invasion depth was confirmed pathologically through specimens obtained after the treatment. This study was approved by the Institutional Review Board of Gangnam Severance Hospital (no. 3-2017-0365).

### 2.2. Data Preparation (Endoscopic Image Collection)

Endoscopy was performed for screening or preoperative examinations. Images were captured using standard endoscopes (GIF-Q260J, GIF-H260, and GIF-H290; Olympus Medical Systems, Co., Ltd., Tokyo, Japan). The image of the lesion should have both close-up and a distant view so that the size and position of the lesion can be identified. Additionally, the amount of gas insufflation should be adjusted appropriately to reflect the condition of the lesion and its surrounding area.

We collected 11,686 endoscopic images, including 1097 T1a-EGC, 1,005 T1b-EGC, and 9834 non-EGC images. The non-EGC images were endoscopic images of the gastric mucosa that were not EGC, including chronic gastritis, chronic atrophic gastritis, intestinal metaplasia, and erosion. The images with poor quality were filtered out. The image inclusion criteria comprised white light images and images with whole lesions. However, images with motion-blurring, out of focus, halation, and poor air insufflation were excluded. Finally, 11,539 images (896 T1a-EGC, 809 T1b-EGC, and 9834 non-EGC) were selected. To prepare the image dataset for the models, the selected images were randomly organized into five different folds to assess how the trained model was generally applicable while avoiding overfitting or testset selection bias [13]. The five folds were used to train and evaluate the deep learning models. All the folds were independent, and the training:validation:testing dataset ratio at each fold was 3:1:1 (Appendix A). The images extracted from one patient were assigned to a fold; therefore, the number of images between the folds differed slightly (Appendix A). The validation set that was a totally independent fold than the training folds was used to observe the training status during the training. After training the model, the other independent fold was used to evaluate the model performance as a testing set. For example, cross validation-group 1 of Appendix A used the first three folds (A, B, and C) as the training set, the fourth fold (D) as the validation set, and the remaining folds (E) as the testing set (Appendix A).

### 2.3. Convolutional Neural Network and Training

Detailed descriptions of the neural network architectures, novel loss function methods, training methods, and algorithms are presented in the Supplement. A short summary is provided below. We used two networks on two training methods to evaluate which one allowed the CNN to be better oriented to EGC regions. The two network models were based on a transfer learning method with the visual geometry group (VGG)-16 network pre-trained on ImageNet, which is a large-scale dataset published for the image classification task to effectively initialize and train network weights [14,15]. The first model was a typical method that computed the loss between the real and predicted classes of input data. The second was a novel method that used the weighted sum of gradient-weighted class activation mapping (Grad-CAM) and cross-entropy losses. To let the model focus on the fine-grained features of EGC regions, we employed a novel loss function by adding Grad-CAM [16,17]. Although most existing visualization methods require an additional module to generate visual explanations, Grad-CAM can visualize activation statuses that are gradually changed over the training time as its initial architectures [18,19]. The gradually activated EGC regions of the input RGB image, which were produced by passing trained layers, are shown in Appendix A. The blue and red colors on Grad-CAM indicate lower and higher activation values, respectively.

The proposed novel method allows the training procedure to optimize an objective that simultaneously minimizes not only the classification error (real classes—predicted classes) but also the localization error (real lesion mask—activated Grad-CAM). The real lesion mask is part of an endoscopic image that the endoscopist identified as real EGC area. We named this novel method “lesion-based VGG-16.” An overview of the proposed algorithm for the computer-aided diagnosis (CAD) of EGC is shown in Appendix A.

### 2.4. Evaluation

To evaluate the performance of EGC detection and depth prediction models, we measured the sensitivity (%), specificity (%), positive predictive value (PPV) (%), negative predictive value (NPV) (%), and area under the curve (AUC) of receiver operating characteristic (ROC) curves by summing all cross-validation folds. Because the number of test images comprising each cross-validation fold was different, it was insufficient to evaluate their generalized performances. Therefore, we randomly selected a fixed number of images from each class of the test dataset. The test set of EGC depth prediction included 300 images, comprising 150 T1a-EGC and 150 T1b-EGC images. The EGC detection model test set included 660 images consisting of 330 EGC and 330 non-EGC images. Additionally, 90 EGC images not included in the cross-validation datasets were also tested. A total of 1590 and 3390 images were evaluated for predicting EGC depth and detecting EGC, respectively.

Since the network was trained by activating EGC regions, activated regions extracted by Grad-CAM at the last convolutional layer could be considered as suspected cancer regions when an endoscopy image was fed to the network. To demonstrate the utility of cases where activated regions can be localized EGC regions, we evaluated the EGC localization performances. Our method for localizing EGC regions from Grad-CAM and evaluation metrics is described in the Appendix A.

### 2.5. Statistical Analysis

Chi-squared and Fisher’s exact tests were used to evaluate the associations among various categorical variables. Univariable and multivariable logistic regression analyses were performed to identify factors significantly affecting the AI accuracy. Odds ratios (ORs) and relevant 95% confidence intervals (CIs) were calculated. Analyses were performed using SAS version 9.4 (SAS Institute, Cary, NC, USA) or IBM SPSS Statistics for Windows, version 23.0 (IBM Co., Armonk, NY, USA) and *p*-values < 0.05 indicated statistical significance.

## 3. Results

### 3.1. Baseline Clinicopathological Characteristics of the Subjects

The patients included 538 men and 262 women with a mean age of 62.6 years (range: 26–92 years). The mean lesion size (±SD) was 23.7 ± 15.1 mm. There were 428 (53.5%) mucosal-depth lesions (T1a-EGC) and 372 (46.5%) submucosal-depth lesions (T1b-EGC). The histology types of lesions according to the World Health Organization (WHO) classification included well-differentiated (321/800, 40.1%), moderately-differentiated (268/800, 33.5%), poorly-differentiated adenocarcinoma (103/800, 12.9%), and signet-ring cell carcinoma (108/800, 13.5%). The histologic types according to the Japanese classification included differentiated (589/800, 73.6%) and undifferentiated adenocarcinoma (211/800, 26.4%). The baseline clinicopathological characteristics of the lesions are summarized in Table 1.

### 3.2. Diagnostic Performance Using the VGG-16

We first tested the VGG-16 trained using the cross-entropy loss function on the selected test image set. The sensitivity, specificity, PPV, NPV, and overall AUC for EGC detection were 80.7%, 92.5%, 91.9%, 82.0%, and 0.938, respectively. The values for EGC depth prediction were 81.7%, 75.4%, 78.0%, 79.3%, and 0.844, respectively (Appendix A).

Subsequently, we evaluated the lesion-based VGG-16 on the same test image set. The sensitivity and specificity for EGC detection were 91.0% and 97.6%, respectively, and the PPV and NPV were 97.5% and 91.1%, respectively. The overall AUC was 0.981. The sensitivity and specificity of the prediction of tumor depth in the lesion-based VGG-16 were 79.2% and 77.8%, respectively, and the PPV and NPV were 79.3% and 77.7%, respectively. The overall AUC was 0.851 (Appendix A and Figure 1).

### 3.3. Localization Ability of the Activated Regions

We compared the localization ability of the activated regions on the last convolutional layer of the VGG-16 with and without using the Grad-CAM loss. Appendix A show the localization performance of the activation results of the EGC detection and depth prediction models, respectively. The correct ratios at the 0.5 overlap ratio for detecting EGCs of the lesion-based VGG-16 and VGG-16 were 0.994 and 0.581, respectively. For predicting EGC depth, the lesion-based VGG-16 correctly activated the EGC regions with a ratio of 0.959 compared to 0.811 for VGG-16. The activated regions extracted from the last convolutional layer of VGG-16 and lesion-based VGG-16 are shown in Appendix A. The activation regions of VGG-16 (the second row of Appendix A) did not precisely cover the actual EGC regions (first two columns of Appendix A), and in some cases, deviated from the EGC regions (last two columns of Appendix A). In contrast, the lesion-based VGG-16 (last row) attempted to completely activate and reach the EGC regions. In depth prediction, lesion-based VGG-16 reflected the actual EGC regions more accurately, as shown in Appendix A. Figure 2 shows some examples of the correctly classified (first two rows) and misclassified (last row) images of the lesion-based VGG-16. Although the model misclassified the presence or depth of EGCs in some cases, the EGC region was accurately activated.

### 3.4. Factors Associated with the Accuracy of Tumor Detection by AI

EGCs with a flat morphology had a significantly lower accuracy for EGC detection than other gross types (*p* = 0.038) (Table 2). Relatively small size (1–13 mm) (*p* = 0.002) and T1a-EGC (*p* = 0.001) were significantly associated with tumor detection. In multivariable analysis, small size (1–13 mm) (*p* = 0.006) and T1a-EGC (*p* = 0.019) showed statistically lower accuracies. The accuracies of EGC detection for tumors ≤5 and ≤10 mm were 88.4% and 89.4%, respectively. The EGC detection did not differ significantly according to the histologic differentiation and location.

### 3.5. Factors Associated with the Accuracy of T-Staging by AI

Undifferentiated-type histology was the only factor significantly associated with a lower accuracy for T-stage prediction in both univariable and multivariable analyses (*p* = 0.001 and 0.033, respectively) (Table 3). The accuracy did not differ significantly according to the size.

The factors associated with T-stage prediction were reanalyzed in undifferentiated-type histology. T1b was only significantly associated with a lower T-stage prediction accuracy (*p* = 0.015) (Table 4). Thus, factors associated with T-staging in undifferentiated-type histology were investigated. Relatively large size (≥14 mm) (*p* = 0.003) and poorly differentiated adenocarcinoma (*p* < 0.001) were significantly associated with T1b in undifferentiated-type histology (Table 5). Among undifferentiated-type EGCs, flat and elevated morphologies were more common in T1a and T1b, respectively.

## 4. Discussion

Although previous studies have reported the clinical efficacy of EUS in T-staging of EGC, the results are conflicting [7,20,21,22]. Some studies have reported that conventional endoscopy is comparable to EUS for the T-staging of EGC [6,23]. Various morphologic features, such as irregular surface and submucosal tumors, like marginal elevation, have been proposed as predictors of tumor invasion depth [24]. Identification and verification of additional morphological features of deep invasion in large datasets would allow a more complete depth prediction.

The sensitivity and overall AUC of EGC detection in the present study were 91.0% and 0.981, respectively, comparable to those in a previous report [10]. The overall AUC of T-staging by our lesion-based VGG-16 system was 0.851, which is higher than that previously reported for EUS prediction [6,22]. Unlike other studies, the present study also analyzed the factors affecting AI diagnosis [10,25]. The diagnostic accuracy of AI for T-staging was significantly affected by histopathologic differentiation. Undifferentiated-type histology was more frequently associated with an incorrect invasion depth diagnosis by AI. By reanalyzing only undifferentiated-type histology, T1b-EGC was significantly associated with an incorrect EGC invasion depth diagnosis by the AI. Interestingly, this finding was similar to that for the analysis in EUS. Previous studies have reported that the accuracy of EUS for depth prediction is poor in undifferentiated-type EGC or T1b-EGC [6,26]. Undifferentiated-type histology and T1b-EGC are two important factors for the decision to perform an extended ER. Therefore, these results can provide important directions for the development of an AI for EGC.

As it is critical that AI is properly trained, we performed extensive experimentation and discussion. There are some challenges in applying the loss function designed to train the classification model for a natural-image dataset to AI for EGC without modification. To overcome these difficulties, we proposed a novel loss function that computed a weighted sum of typical classification and Grad-CAM losses. By applying the proposed loss function to EGC detection and EGC depth prediction models, the optimizer simultaneously minimized classification and localization losses in the activated Grad-CAM regions. Although there was no significant performance improvement in predicting the depth of EGCs between VGG-16 (AUC = 0.844) and lesion-based VGG-16 (AUC = 0.851), the trained lesion-based VGG-16 predicted the depth of EGCs by automatically activating EGC regions, whereas VGG-16 did not. The classification performance of VGG-16 trained by cross-entropy loss alone is still debatable regarding dataset bias, where the model considered non-EGC regions to optimize the objective. To the best of our knowledge, this is the first study to use a novel loss function that allows the optimizer to determine an optimum by simultaneously considering EGC depth prediction and localization losses of the activated regions. This model uses the proposed method to simultaneously provide prediction and localization.

To determine which proposed loss function made the CNN focus on the EGC region regardless of the network, we trained an 18-layer residual network (ResNet-18) as a CNN-based EGC depth prediction model [27]. We fine-tuned all weights for a ResNet-18 pre-trained on the ImageNet Dataset. The activation results of ResNet-18 are shown in Appendix A. As with VGG-16, ResNet-18 was also trained using two types of loss functions [27]. As shown in the last two columns of Appendix A, the lesion-based ResNet-18 more accurately activated the EGCs as compared to ResNet-18.

The present study has several limitations. First, we did not analyze the accuracy of EGC detection according to the background mucosa. That is, the background mucosa of the stomach is accompanied by chronic inflammatory changes such as chronic atrophic gastritis and intestinal metaplasia. These are important features that complicate EGC diagnosis. However, to overcome the differences in accuracy based on the features of the background mucosa, more than 9800 non-EGC endoscopic images were learned. Second, the number of undifferentiated-type histology cases was relatively smaller than that of differentiated-type histology. The AI performance is related to the amount of data, and thus may have played an important role in the accurate prediction of EGC depth. It is possible that growth patterns or biological characteristics of undifferentiated histology areas are also affected. Similar findings were reported in previous EUS studies. Third, we did not compare the diagnostic accuracy of lesion-based VGG-16 to that of endoscopists for all images of the study, although endoscopists predicted the invasion depth for subsets of images in this study, with a sensitivity of 76% and overall accuracy of 73% (data not shown). Therefore, the proposed method may be a good tool for predicting the depth of EGC invasion. Finally, this is a retrospective study. Standardization of images is a very important part of the research involving image analysis. The images used were of good quality, and they appropriately characterized the lesions. However, they were not completely standardized with numerical analysis. To overcome the aforementioned limitations, we plan to perform research by using endoscopic video in the future.

## 5. Conclusions

In conclusion, AI may be a good tool for not only EGC diagnosis but also for the prediction of invasion depth, especially in differentiated-type EGC. To maximize the clinical usefulness, it is important to choose an appropriate method for AI application. The lesion-based model is the most appropriate training method for AI in EGC. EGC with undifferentiated-type histology and T1b-EGC is more frequently associated with an incorrect EGC invasion depth by AI. The development of a well-trained AI for undifferentiated-type histology and T1b-EGC is warranted. Further study is also necessary to understand the operating principles of AI and to validate these findings.

## Figures and Tables

**Figure 1 jcm-08-01310-f001:**
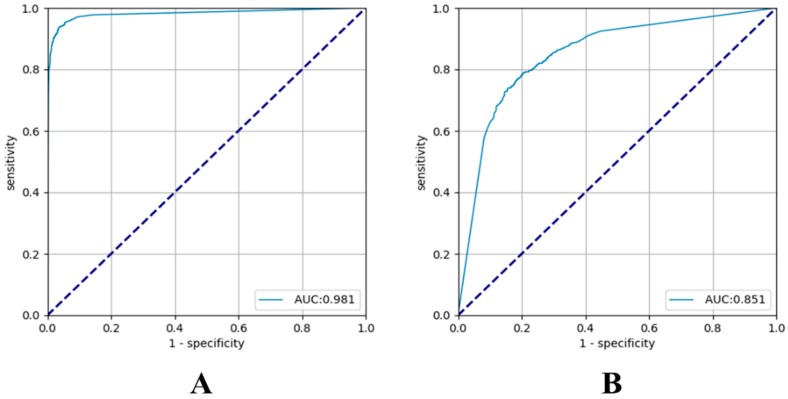
Receiver operating characteristic (ROC) curves of lesion-based visual geometry group (VGG)-16 for the test dataset with their areas under the curves (AUCs). (**A**) Early gastric cancer (EGC) detection model and (**B**) EGC depth prediction model.

**Figure 2 jcm-08-01310-f002:**
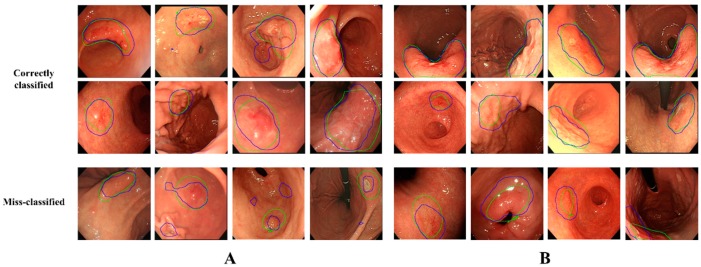
Classification results of lesion-based VGG-16. The green lines indicate the actual early gastric cancer (EGC) regions. The blue lines indicate the activated regions at testing. The first two rows are images precisely classified to their own classes, whereas the last row shows misclassified images. (**A**) EGC detection. (**B**) EGC depth prediction.

**Table 1 jcm-08-01310-t001:** Baseline clinicopathological characteristics of all patients.

Characteristics	Value
Age (years, mean ± SD)	62.6 ± 12.2
Male (*n*, %)	536 (67.2)
Tumor size (mm, mean ± SD)	23.7 ± 15.1
Location of lesion (*n*, %)	-
Upper one-third	74 (9.3)
Middle one-third	118 (14.7)
Lower one-third	608 (76)
Gross type (*n*, %)	-
Elevated	171 (21.4)
Flat	285 (35.6)
Depressed	344 (43)
Lymphovascular invasion (*n*, %)	82 (10.3)
Perineural invasion (*n*, %)	14 (1.8)
T-stage (*n*, %)	-
Mucosa (T1a)	428 (53.5)
Submucosa (T1b)	372 (46.5)
WHO classification (*n*, %)	-
Well-differentiated	321 (40.1)
Moderately-differentiated	268 (33.5)
Poorly-differentiated	103 (12.9)
Signet ring cell carcinoma	108 (13.5)
Japanese classification (*n*, %)	-
Differentiated	589 (73.6)
Undifferentiated	211 (26.4)
Lauren classification (*n*, %)	-
Intestinal	606 (77.3)
Diffuse	156 (19.9)
Mixed	22 (2.8)

**Table 2 jcm-08-01310-t002:** Factors affecting the accuracy of tumor detection.

Variables	Accurate	Inaccurate	*p*-Value	Odds Ratio (95% CI)	*p*-Value
Gross type (*n*, %)	-	-	0.038	-	-
Elevated	169 (21.7)	2 (10.5)	-	-	-
Flat	271 (34.9)	12 (63.2)	-	-	-
Depressed	337 (43.4)	5 (26.3)	-	-	-
T-stage (*n*, %)	-	-	0.001	-	0.019
Mucosa (T1a)	406 (52.3)	17 (89.5)	-	ref	-
Submucosa (T1b)	371 (47.7)	2 (10.5)	-	5.891 (1.326–26.171)	-
Size (*n*, %)	-	-	0.002	-	0.006
1–13 mm	162 (21.7)	11 (57.9)	-	ref	-
≥14 mm	608 (78.3)	8 (42.1)	-	3.660 (1.427–9.384)	-
Location of lesion (*n*, %)	-	-	0.780	-	-
Upper one-third	72 (9.3)	2 (10.5)	-	-	-
Mid one-third	115 (14.8)	3 (15.8)	-	-	-
Lower one-third	590 (75.9)	14 (73.7)	-	-	-
Japanese classification (*n*, %)	-	-	0.296	-	-
Differentiated	575 (74)	12 (63.2)	-	-	-
Undifferentiated	202 (26)	7 (36.8)	-	-	-

**Table 3 jcm-08-01310-t003:** Factors affecting the accuracy of T-staging.

Variables	Accurate	Inaccurate	*p*-Value	Odds Ratio (95% CI)	*p*-Value
Japanese classification (*n*, %)	-	-	0.001	-	0.033
Differentiated	446 (76.8)	132 (65.0)	-	ref	-
Undifferentiated	135 (23.2)	71 (35.0)	-	0.491 (0.255–0.945)	-
Gross type (*n*, %)	-	-	0.442	-	-
Elevated	127 (21.9)	41 (20.2)	-	-	-
Flat	212 (36.5)	67 (33.0)	-	-	-
Depressed	242 (41.6)	95 (46.8)	-	-	-
T-stage (*n*, %)	-	-	0.235	-	-
Mucosa (T1a)	320 (55.1)	102 (50.3)	-	-	-
Submucosa (T1b)	261 (44.9)	101 (49.7)	-	-	-
Size (*n*, %)	-	-	0.329	-	-
1–13 mm	137 (23.7)	44 (21.8)	-	-	-
≥14 mm	442 (76.3)	158 (78.2)	-	-	-

**Table 4 jcm-08-01310-t004:** Factors affecting the accuracy of T-staging in undifferentiated-type adenocarcinoma.

Variables	Accurate	Inaccurate	*p*-Value	Odds Ratio (95% CI)	*p*-Value
T-stage (*n*, %)	-	-	0.015	-	0.015
Mucosa (T1a)	97 (71.9)	39 (54.9)	-	ref	-
Submucosa (T1b)	38 (28.1)	32 (45.1)	-	0.477 (0.262–0.869)	-
Gross type (*n*, %)	-	-	0.152	-	-
Elevated	15 (11.1)	7 (9.9)	-	-	-
Flat	55 (40.7)	20 (28.1)	-	-	-
Depressed	65 (48.2)	44 (62.0)	-	-	-
Size (*n*, %)	-	-	0.444	-	-
1–13 mm	24 (17.9)	14 (19.7)	-	-	-
≥ 14 mm	110 (82.1)	57 (80.3)	-	-	-
WHO classification (*n*, %)	-	-	0.296	-	-
APD	60 (44.4)	38 (53.5)	-	-	-
SRC	75 (55.6)	33 (46.5)	-	-	-

APD, poorly differentiated adenocarcinoma; *SRC*, signet ring cell carcinoma.

**Table 5 jcm-08-01310-t005:** Associated factors according to T-staging in undifferentiated-type adenocarcinoma.

Variables	T1a	T1b	*p*-Value
Gross type (*n*, %)	-	-	0.003
Elevated	8 (5.8)	14 (19.2)	-
Flat	57 (41.3)	19 (26.0)	-
Depressed	73 (52.9)	40 (54.8)	-
Sex (*n*, %)	-	-	0.012
Male	60 (43.5)	45 (61.6)	-
Female	78 (56.5)	28 (38.4)	-
Size (*n*, %)	-	-	0.003
1–13 mm	33 (24.1)	6 (8.2)	-
≥14 mm	104 (75.9)	67 (91.8)	-
Location of lesion (*n*, %)	-	-	0.276
Upper one-third	5 (3.6)	5 (6.8)	-
Mid one-third	27 (19.6)	19 (26.0)	-
Lower one-third	106 (76.8)	49 (67.1)	-
WHO classification (*n*, %)	-	-	<0.001
APD	53 (38.4)	50 (68.5)	-
SRC	85 (61.6)	23 (31.5)	-

APD, poorly differentiated adenocarcinoma; *SRC*, signet ring cell carcinoma.

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
