# Peer review of "A Lesion-Based Convolutional Neural Network Improves Endoscopic Detection and Depth Prediction of Early Gastric Cancer"

_jcm, 2019, doi:10.3390/jcm8091310_

Round 1

Reviewer 1 Report

The authors describe in this work an attempt to recognize gastric cancerous lesions and deep invasion from endoscopic photographs. Thousands of white light images were incorporated in a data base in order set up an automatized recognition of cancer and deep invasion.

Two major criticisms off the methods limit the scope of the findings

1 As well known by the endoscopists the characterisation of the endoscopic lesions depends of many conditions (light, sharpness of the images, proximity of the lens, angle of vison, use of high definition endoscope). As far the reader is able to know from the method part and the illustrations none of these parameters has been standardised. It is probably due the retrospective use of images from a collection.

As the image source is not standardised it seems difficult to be confident with the analysis results.

2 In this type of work is it mandatory to validate the detection process on an independent set of images. This work remains to be performed.

Il is useless to conclude that the automatic recognition of lesion is successfully working on the same data base that those used for machine learning. 

Author Response

We are thankful for the thorough review of our paper.

Two major criticisms off the methods limit the scope of the findings

1. As well known by the endoscopists the characterisation of the endoscopic lesions depends of many conditions (light, sharpness of the images, proximity of the lens, angle of vison, use of high definition endoscope). As far the reader is able to know from the method part and the illustrations none of these parameters has been standardised. It is probably due the retrospective use of images from a collection.

As the image source is not standardised it seems difficult to be confident with the analysis results.

1.(1) Thank you for your valuable comment. We agree with your observation regarding the possibility of lack of standardization owing to the retrospective nature of the study. However, we have carefully performed the image selection process.

As mentioned in the manuscript, the images were captured using standard endoscopes (GIF-Q260J, GIF-H260, and GIF-H290; Olympus Medical Systems, Co., Ltd., Tokyo, Japan).

In normal clinical practice, the endoscopic diagnosis is done by observing the lesion at various angles and distances. Therefore, the images of the lesion should have both a close-up and a distant view so that the size and position of the lesion can be identified. Additionally, the amount of gas insufflation should be adjusted appropriately to reflect the condition of the lesion and its surrounding area. All the images were captured by considering of these conditions. However, the possibility of improper images owing to technical defects or blunders cannot be ruled out. Therefore, we performed filtering to satisfy the following criteria during the image collection process. The image inclusion criteria comprised white light images and images with whole lesions. The exclusion criteria were any images that were poor quality images resulting from motion-blurring, out of focus, halation, and poor air insufflation.

We modified the description in the method section to explain this image selection process as follows:

(2) All our images meet these criteria, although they are not numerically standardized. Therefore, there is a possibility of eliminating all the bias on the general retrospective study. Even if a deep learning model is trained on absolutely clean and noise-controlled images, the model cannot perform perfectly on new input images in the wild. Training diverse images that do not disturb the pattern of the image dataset in some portion makes the model more generalized. Consequently, we preferentially filtered the images interrupting what the model learns about the patterns of the dataset. Although setting up the specific portion of noisy images on the dataset according to parameterized standards is ideal, it has to be researched by performing in-depth analysis of the endoscopic images, which comprises our future works. Standardization of images is an important part of research involving image analysis. Consequently, we added the following sentences to the discussion section as follows:

(1) Methods

2.2 Data preparation (endoscopic image collection)

Endoscopy was performed for screening or preoperative examinations. Images were captured using standard endoscopes (GIF-Q260J, GIF-H260, and GIF-H290; Olympus Medical Systems, Co., Ltd., Tokyo, Japan). The image of the lesion should have both close-up and a distant view so that the size and position of the lesion can be identified. Additionally, the amount of gas insufflation should be adjusted appropriately to reflect the condition of the lesion and its surrounding area.

We collected a total of 11,686 endoscopic images, including 1,097 T1a-EGC, 1,005 T1b-EGC, and 9,834 non-EGC images. The non-EGC images were endoscopic images of the gastric mucosa that were not EGC, including chronic gastritis, chronic atrophic gastritis, intestinal metaplasia, erosion, etc. Poor-quality images were excluded, including those out of focus, with an inadequate angle to view the lesion, and with poor air insufflation.

The images with poor quality were filtered out. The image inclusion criteria comprised white light images and images with whole lesions. However, images with motion-blurring, out of focus, halation, and poor air insufflation were excluded.

(2) Discussion

-------- The present study has several limitations. ------- Finally, this is a retrospective study. Standardization of images is a very important part of the research involving image analysis. The images used were of good quality, and they appropriately characterized the lesions. However, they were not completely standardized with numerical analysis. To overcome the aforementioned limitations, we plan to perform research by using endoscopic video in the future.

2. In this type of work is it mandatory to validate the detection process on an independent set of images. This work remains to be performed.

Il is useless to conclude that the automatic recognition of lesion is successfully working on the same data base that those used for machine learning.

2. Thank you for your valuable comment. The cross validation technique is often used in artificial intelligence (AI) learning and testing using images. Learning, validation, and testing are conducted on independent data (lesions). Previously, this technique was introduced in an article on AI learning using colonoscopy images [Gastroenterology 2018;155:1069–1078]. All the folds were independent of each other and the training:validation:testing dataset ratio at each fold was 3:1:1. The validation set that was a totally independent fold than the training folds was used to observe the training status during the training. After training the model, the other independent fold was used to evaluate the model performance as a testing set. For example, group 1 of supplementary Table 1 (newly added) uses the first three folds (A,B and C) as the training set, the fourth fold (D) as the validation set, and the remaining folds (E) as the testing set. The configurations of these datasets comprise an important part of the research. We modified the method section and added the below table as supplementary information to improve the reader’s understanding as follows:

Supplementary Table 1. Groups for 5-fold cross validation

Group

Training

Validation

Test

1

A,B,C

D

E

2

B,C,D

E

A

3

C,D,E

A

B

4

D,E,A

B

C

5

E,A,B

C

D

2.2 Data preparation (endoscopic image collection)

--------------------------------------------------------------------------Finally, 11,539 images (896 T1a-EGC, 809 T1b-EGC, and 9834 non-EGC) were selected. To prepare the image dataset for the models, 11539 images were randomly organized into five different folds to assess how the trained model was generally applicable while avoiding overfitting or test set selection bias.13 The five folds were used to train and evaluate the deep learning models. All the folds were independent, and the training:validation:testing dataset ratio at each fold was 3:1:1 (Supplementary Table 1). The images extracted from one patient were assigned to a fold, the number of images between folds differed slightly (Supplementary Table 1 2). The validation set that was a totally independent fold than the training folds was used to observe the training status during the training. After training the model, the other independent fold was used to evaluate the model performance as a testing set. For example, cross validation-group 1 of supplementary Table 1 used the first three folds (A, B and C) as the training set, the fourth fold (D) as the validation set, and the remaining folds (E) as the testing set (Supplementary Table 1).

Reviewer 2 Report

This paper contains well-designed research on how to employ a convolutional neural network for early gastric cancer staging to benefit the treatment plan (and, hopefully, the patient's prognosis). The main limitations of the study are outlined in the Discussion. The statistical analyses seem sound and well-applied. Suggestions for improvements:

- please have a (native) English professional check the English spelling and grammar. Especially in the introduction and methods sections, improvements can be made.

-  the undifferentiated-type is underrepresented in the training set. However, I noticed that the ratio between male and female patients is unequal, too. Can you please comment on that?

Author Response

For reviewer 2:

Thank you for the insightful review of our paper.

(1) Please have a (native) English professional check the English spelling and grammar. Especially in the introduction and methods sections, improvements can be made.

Answer>

(1) Thank you for your valuable feedback. According to your suggestion, we had our manuscript reviewed by a professional English editing service (Editage).

(2) The undifferentiated-type is underrepresented in the training set. However, I noticed that the ratio between male and female patients is unequal, too. Can you please comment on that?

Answer>

(2) Thank you for your keen observation. The incidence of gastric cancer in Korea is higher in males than females as explained hereafter. Age-adjusted annual incidence of gastric cancer per 100,000 persons is 62.8 for males and 25.7 for females [GLOBOCAN 2012 v1.0]. However, in our study, the incidence for males is 67.2% that is slightly higher than prevailing distribution in Korea.

The reason for this is probably the different portion of histologic differentiation such as differentiated and undifferentiated-type gastric cancer in our study. Intestinal type or differentiated-type gastric cancer is more common in males, whereas diffuse type or undifferentiated-type gastric cancer is equally seen in both sexes. [Br J Cancer 1995 Aug; 72(2): 424-426], [Int J Cancer 1968 Nov 15;3(6):809-18]. Therefore, we considered that the small proportion of undifferentiated-type gastric cancer in our study affected the sex distribution.
